# Melding Fog Computing and IoT for Deploying Secure, Response-Capable Healthcare Services in 5G and Beyond

**DOI:** 10.3390/s22093375

**Published:** 2022-04-28

**Authors:** Christos Tselios, Ilias Politis, Dimitrios Amaxilatis, Orestis Akrivopoulos, Ioannis Chatzigiannakis, Spyros Panagiotakis, Evangelos K. Markakis

**Affiliations:** 1Wireless Communications Laboratory, University of Patras, 26504 Patras, Greece; 2Systems Security Laboratory, University of Piraeus, 18532 Piraeus, Greece; ipolitis@ssl.ds.unipi.gr; 3InQbit Innovations SRL, Peroni Str. District 4, No. 14–16, 041386 Bucharest, Romania; 4Spark Works Limited, Unit 2B, Galway Technology Centre, Mervue Business & Technology Park, Wellpark Road, H91 AHR1 Galway, Ireland; d.amaxilatis@sparkworks.net (D.A.); akribopo@sparkworks.net (O.A.); 5Department of Computer, Control and Informatics Engineering, La Sapienza University, 00185 Rome, Italy; ichatz@diag.uniroma1.it; 6Department of Electrical and Computer Engineering, Hellenic Mediterranean University, 71500 Heraklion, Greece; spanag@hmu.gr

**Keywords:** Fog Computing, Internet of Things, B5G, cybersecurity, healthcare services

## Abstract

The fifth generation (5G) of mobile networks is designed to mark the beginning of the hyper-connected society through a broad set of novel features and disruptive characteristics, delivering massive connectivity, coverage and availability paired with unprecedented speed, throughput and capacity. Such a highly capable networking paradigm, facilitated by its integrated segments and available subsystems, will propel numerous cutting-edge, innovative and versatile services, spanning every possible business vertical. Augmented, response-capable healthcare services have already been identified as one of the prime objectives of both vendors and customers; therefore, addressing controversies and shortcomings related to the specific field is considered a priority for all stakeholders. The scope of this paper is to present the architectural elements of 5G which enable efficient, remote healthcare services along with emergency health monitoring and response capability. In addition, we propose a holistic scheme based on technical enablers such as Internet-of-Things (IoT) and Fog Computing, for mitigating common issues and current limitations which may compromise the proclaimed service delivery.

## 1. Introduction

Legacy telecommunication networks remain, to this day, the core for widely used e-health monitoring and emergency response solutions, despite their inability to cope with the agile, robust, highly-available infrastructure or the rich-media-content devices users are currently relying on. On the other hand, smartphones and their inherent ability of sharing multimodal information has led to a constant adoption increase by all consumers which aim to augment their overall mobile quality of experience through a plethora of diverse, yet fully functional voice, video, text messaging and social media networking applications. This trend raised the total percentage of emergency calls made by mobile devices to almost 70% out of the 300 million of such calls in the European Union annually [1], a clear indication that mobile equipment, facilitated by the constant improvement of the underlying infrastructure, will most likely eradicate all other types of emergency communication appliances. Moreover, the new capabilities provided by these new devices, spanning from conveying precise location information to real-time metrics and multimedia content from the incident, significantly increase the chances of a successful emergency response or an accurate remote diagnosis, factors which can sometimes make the difference between life and death.

The fifth generation (5G) of mobile networks is designed to mark the beginning of the hyper-connected society, by delivering significant performance improvement paired with an extraordinary set of new features and disruptive characteristics. The imminent deployment of 5G networking infrastructure is expected to offer massive connectivity, coverage and availability, extremely high data rate, less than 1 ms end-to-end over-the-air latency, significantly larger capacity and throughput, real-time information processing and data transmission along with native support for diverse verticals, services and applications, each with different specification and requirements [2]. In regards to response-capable healthcare solutions, improved device-to-device communication will expand the uplink capacity and availability of communication channels, creating an always connected experience for patients, healthcare personnel and emergency workers, which will now have improved awareness through access to high-quality multimedia content [3]. The novel feature of network-slicing introduced by 5G, is expected to facilitate dynamic adaptation of both transmission speed and network latency, thus ensuring high priority for emergency communications as a whole.

However, in accordance to 5G, there are certain contemporary frameworks and networking paradigms that will act as technical enablers for further amplifying the capabilities of existing e-health and emergency response platforms to a whole new level. The Internet-of-Things (IoT), a highly capable ecosystem that will allow the deployment of networking topologies of unprecedented scale, as well as Fog Computing [4], a multi-tier architecture for delivering an efficient, highly virtualized platform that provides resources for computational, networking and storage services on low-capability nodes located at the edge of the network yet in close cooperation with traditional cloud servers at the network’s backend infrastructure, will redefine healthcare service capabilities as well as the ability of emergency response by all participating entities.

In this article, we discuss how response-capable healthcare services can benefit from the upcoming telecommunication ecosystem disruption, and present a novel, cutting-edge architecture which integrates new services on top of existing core functionality, thus adding significant value through the use cases it supports. Delivering a holistic, end-to-end framework for interconnecting all major health-case service stakeholders is the ultimate goal, and building such a solution by taking advantage of key architectural elements of the 5G ecosystem is as radical as it gets. Section 2 presents the technical enablers for contemporary health monitoring services, while Section 3 lists certain principles and use cases which the design of such services must follow or address. The proposed system architecture blueprints are described in Section 4 and, finally, Section 5 summarizes results and concludes the paper.

## 2. Technical Enablers for Contemporary Health-Monitoring Services

### 2.1. IoT and Fog Computing

The main concept of IoT revolves around the deployment of abundant network topologies of interconnected sensors and actuators. Equipped with unique identifiers and transmission mechanisms, everyday objects will be capable of automatically utilizing affiliated interfaces and gateways to upload vast amounts of diverse data. IoT extends internet connectivity beyond traditional devices to a new range of nodes now able to connect, interact and exchange information. The now-upgraded abilities of previously simplistic sensing devices will accelerate the wider adoption of wearable low-power IoT medical sensors for monitoring health-related data and will consequently lead to the development of a digitized healthcare system, able to directly link patients with the available medical resources and corresponding services. However, certain limitations due to existing infrastructure inefficiency do exist, and can only be tackled by harnessing the benefits of a 5G-based architecture, similar to the one proposed in Section 4 in terms of speed, capacity, connectivity and efficiency.

All contemporary IoT platforms appear to follow a specific four-stage data-handling pattern. In the first stage, edge sensing devices collect real-world metrics and convert continuous-time signals to discrete-time ones using efficient sampling techniques. The second stage involves data accumulation and transfer to interconnected facilities in the cloud for further processing. The third stage contains the actual data analysis conducted by dedicated equipment while the fourth one incorporates processed information forwarding back to the edge for triggering actuators. It becomes obvious that the specific data processing pipeline consumes bandwidth and network resources in general, which become more scarce as the overall topology size increases.

As stated in [5], minimal end-to-end latency together with network bandwidth preservation as well as enhanced reliability are considered attributes of significant importance for any IoT-related application. Moreover, healthcare applications also introduce a need for robust data collection, storage and availability mechanisms, paired with a demand for uninterrupted services even in cases of compromised cloud connectivity and resource constrained devices [6,7]. When these restrictions are combined with the prerequisites for near-real-time data processing [8] and enhanced data security and privacy [9,10,11,12], they form an amalgam of challenges where only radical, end-to-end solutions [13] may stand a chance.

Fog and edge computing in general is a rising, highly virtualized platform which complements IoT by providing computational, networking and storage services in an intermediate layer between end-user equipment and conventional cloud datacenters [4]. Fog infrastructure most of the time is deployed in physical proximity with the last-mile sensors of IoT-related applications. These sensors may be resource-bound by design yet the interaction of the two introduces significant reliability and security improvement, limits bandwidth consumption and greatly decreases end-to-end latency. When deployed in real-world use cases, the specific approach extends traditional distributed computing paradigms by transferring data processing closer to the production sites, thus accelerating the frameworks’ responsiveness to triggering events along with its awareness, after eliminating the necessity for data transfer to remote processing nodes [14,15]. Besides improvements in network delay, Fog Computing has the potential for providing additional services, including: (i) improved location awareness; (ii) seamless device mobility and inter-domain roaming; (iii) enhanced device heterogeneity; and (iv) augmented scalability in terms of concurrent interconnected nodes. As stated in [14], the specific solution tackles the majority of the inherent limitations of the cloud and facilitates service deployment with near-zero error tolerance, such as industrial and healthcare ones. Harnessing the benefits of Fog infrastructure deployment is especially critical for healthcare-related use cases, since timely response to a signal coming from a sensor connected to a patient can make the difference between life and death.

### 2.2. 5G

The advent of 5G introduced a series of integral changes to already-deployed infrastructure, with the most important being the conversion of virtually every service into software function [16]. This evolution will thrust telecommunications into a new era, dominated by smart cities, factories and grids, autonomous vehicles and massive IoT framework deployments delivering solutions spanning from healthcare to mechanized agriculture. The vertical application range is so wide that can be only broadly classified into three categories, namely: (i) enhanced mobile broadband (eMBB); (ii) ultra-reliable and low-latency communications (URLLC); and (iii) massive machine-type communications (mMTC). eMBB traffic is characterized by large payloads and considered to be an extension of the existing 4G broadband service, but with improved performance and superior overall user experience. Its objective is to maximize data rate while guaranteeing a certain degree of connection reliability paired with low-pace mobility, coverage and seamless connectivity on densely populated areas and hotspots. URLLC on the other hand has much stringent reliability, availability and latency requirements, therefore such transmissions should be kept localized and with a relatively low rate. When reliability becomes the top priority, resource provisioning may become challenging, thus in need of a dynamic network readjustment technique for avoiding idle or misused elements. mMTC traffic patterns are not yet fully identified but will most likely consist of fixed, typically low, uplink transmission rates. However, since mMTC deployments will incorporate a very large number of devices, even transmitting volumes of non-delay-sensitive data may require random-access resource management schemes, thus making the design objective of mMTC to maximize the arrival rate in the available radio resources. The technical enablers of all aforementioned application types are listed in the following paragraphs.

#### 2.2.1. SDN/NFV

Software-Defined Networking is the dominant architectural framework for deploying highly intelligent, agile and easily programmable networking topologies. The main novelty it introduces lies in the decoupling of all control and data-forwarding entities, an approach which disrupts vertical integration and consequently separates the core network control mechanism from all underlying routing and switching elements, with the latter being deprived of any perplexed functionality left only to operate as packet forwarders. The overall concept also involves the existence of a centralized software controller that maintains a global network overview through integrated control functions, as well as being responsible for generating or modifying the necessary forwarding policies before these are being distributed to the edges of the network. This approach allows seamless and more dynamic network topology adjustments by eliminating the need for independently accessing and reconfiguring scattered devices, often located in different geographical points of presence.

The second pillar of efficient network topology deployment is the Network Function Virtualization (NFV) paradigm which intends to convert physical appliances into highly adaptive software and replace proprietary hardware devices with commercial off-the-shelf (COTS) general-purpose equipment. One of the most thorough architectural framework propositions for NFV was introduced by the ETSI NFV Industry Specification Group (ISG) containing the necessary building blocks, communication interfaces and generic concepts for assembling and operating such a perplexed, interconnected and distributed ecosystem. According to ETSI, any NFV architecture integrates three key elements: the Virtual Network Functions (VNF), the Network Function Virtualization Infrastructure (NFVI) and NFV Management and Orchestration (NFV MANO) [17]. VNFs are considered to be the prime functional blocks of the networking infrastructure, NFVI resembles to the underlying resources which generate the operating environment for the VNFs, while the NFV MANO overlooks VNF provisioning and controls all operational tasks. Following the NFV paradigm, any provided service will be divided into several VNFs whose nature, amount and order will be determined by their functional and behavioral specifications, thus playing the role of the service’s structural elements, accountable for every aspect of its behavior in an end-to-end manner. The affinity of NFV and SDN now becomes more obvious. Both solutions appear to advocate towards the replacement of existing networking elements with open-source software operating on top of generic and universally deployed hardware platforms. Moreover, they both heavily rely on virtualization to support their seamless functionality and deliver an extensive feature set, leading many researchers and practitioners to consider them complementary. This notion is also boosted by the commonality of having an NFV deployment integrate an SDN controller in its value chain, thus taking advantage the inherent reliability and elasticity features it incorporates [2]. The delivery of healthcare-related applications is not directly benefited by SDN and NFV; however, these two revolutionary networking approaches create the bedrock on which all modern 5G applications are built. It is therefore crucial to design and deploy a framework that fully supports them, such as the one presented in Section 4, to inherently increase their speed, resilience and scalability.

#### 2.2.2. Network Slicing

Network slicing relies on utilizing virtualization technology through paradigms such as SDN or NFV to segregate communication and computational resources available on restricted physical infrastructure towards generating multiple isolated logical networks, each optimized to support a specific service delivery. Figure 1 illustrates the core concept of network slicing with finite network resources partitioned into multiple virtual segments (or slices) each with their own architecture, application provisioning, packet and signal-processing capacity, targeting a specific number of end users [18,19].

Any network slicing deployment must support two distinct states: creation and runtime [20]. During the slice-creation state, a valid service request issued by the end user in accordance with a predefined network service inventory triggers the slice-creation process which must be integrated to the network operator infrastructure. In the runtime state, user requests are accommodated by already-allocated resources shaped into a slice. Slicing renders operators capable of providing new services and applications only by activating a fraction of their existing infrastructure instead of rolling out a new network, an approach which reduces capital expenditure and deployment time while simplifying the creation, configuration and operation of network services in general [21,22,23,24,25].

5G communication networks will support two slicing dimensions: (i) vertical, a segmentation of the core mobile network domain based on predefined applications; and (ii) horizontal, which is designed to accommodate new trends for scaling system capacity, enable cloud computing cooperation and support computation offloading to the network edge. Real-world healthcare applications are essentially handicapped without network slicing, since the ability to divide network resources into isolated chunks ensures that under no circumstances will a sensor transmitting patient-originating data lack the necessary bandwidth, throughput, capacity and connectivity to successfully deliver life-critical information to the correct recipient, namely a doctor, a medical facility or an incoming ambulance. The proposed architecture presented in Section 4 is based on the assumption that resource scarcity is not an issue, and this exact assumption became a reality only after the advent of 5G Networking and the abundance of features it introduced.

## 3. Designing Response-Capable Healthcare Application Frameworks

### 3.1. Next-Generation Emergency Services

Emergency response services as provided by operators need a radical transformation to align with the next-generation networking blueprints currently defined by standardization bodies [26]. Legacy systems based on Public Safety Answering Points (PSAPs) already struggle to support contemporary service demands; therefore, all corresponding regulatory authorities such as the National Emergency Number Association (NENA) and the European Emergency Number Association (EENA) envision an all-IP based network for emergency service forwarding, currently manifested by NG112 architecture. In this framework, caller location is of paramount importance since it enables proper call routing to the most convenient emergency agency. A dedicated entity, the Emergency Call Routing Function (ECRF) using the Location-to-Service Translation Protocol (LoST) handles the routing, facilitated by the Emergency Services Routing Proxy (ESRP). ESRP is also responsible for (i) evaluating a policy rule set for the queue a call arrives onto, (ii) determine the most appropriate next hop URI via ECRF query and (iii) evaluate a policy rule set for the aforementioned URI based on auxiliary information such as message content, time of day or PSAP state [27,28,29,30].

Under the auspices of EMYNOS Project (https://www.emynos.eu/ accessed on 12 February 2022) extended end-to-end measurements were conducted over a carefully designed testbed, which mimics the behaviour of a state-of-the-art emergency communication platform. More specifically, we used a Linphone (https://www.linphone.org/ accessed on 12 February 2022) generic interface on the client side to initiate specific SIP message exchange with the PSAP, in an attempt to stress-test and validate the ability of the infrastructure to parse arriving sensor values. Such testing will reveal possible limitations of the emergency communication platform SIP core (ECRF, ESRP) on handling inbound SIP traffic, mostly containing SUBSCRIBE/NOTIFY messages, under load. In all testing cases, SIPp (http://sipp.sourceforge.net/ accessed on 12 February 2022) , a free Open Source traffic generator for the SIP protocol was used for stressing the SIP protocol’s behavior while NetEm (https://wiki.linuxfoundation.org/networking/netem accessed on 12 February 2022) was used for emulating diverse networking behavior [3].

### 3.2. Experimental Evaluation Based on Next-Generation Emergency Services

During the structural testing preparatory phase, we heavily modified the Linphone software bundle that was later deployed on the client side. Modifications included implementing a generic SIP-based interface, able to transport SensorML-formatted data, thus providing seamless communication with a large variety of sensor platforms. This newly implemented interface uses the SIP MESSAGE method to allow information/data flow across every sensor platform that is considered an emergency event actuator. Determining the interface’s availability while attempting to receive/broadcast messages to multiple interconnected platforms constitutes the main scenario of the proposed functional test. SIPp played the role of the actuator, creating random amounts of SIP MESSAGES, while Linphone primarily converted inbound SIP MESSAGES to corresponding SIP NOTIFY ones. The testing sequence was executed for 10 times, creating a total of 100 SIP MESSAGES in each run with a variable rate between 30 and 60 messages per second. As shown in Table 1, when the rate was set to 30 SIP MESSAGES per second, the interface failed to properly handle 57 out of 1000 SIP MESSAGES on average. Moreover, the average number of unacknowledged SIP MESSAGES for the various rates together with the corresponding retransmission number per case is also depicted. It must be stated that SIPp configuration for the experiment, needs to be identical to the generic SIP retransmission mechanism, mandating the retransmission time (T1) to be configured at 500 ms and the timeout timer (TF) to 4 s.

PSAP efficiency and robustness on parsing incoming SensorML formatted data was evaluated through a separate functionality test, which identified the number of improperly handled (unacknowledged) SIP NOTIFY messages on the corresponding SIP client. Similar to the previous testing sequence, there was a total of 10 consecutive runs each with 1000 and 2000 SIP NOTIFY messages that were transmitted to the PSAP with various rates. It is evident that there is no correlation to the behaviour of SIP NOTIFY messages with the overall number of transmitted messages, while in the same time there is an almost linear increase of the unacknowledged SIP NOTIFY messages number against the overall number of exchanged messages [3].

Table 2 contains the average number of unacknowledged SIP NOTIFY messages for the various rates of the specific sequence, paired with the corresponding retransmission number. As previously, SIPp configuration for the experiment needs to be identical to the generic SIP retransmission mechanism, mandating the retransmission time (T1) to be configured at 500 ms and the timeout timer (TF) to 4 s.

For conducting a realistic functionality testing, we should take into consideration that the EMYNOS Prototype utilizes sensor platforms as actuators for automatically triggered emergency calls. The sensor data are delivered to the PSAP side as a mean to enhance the overall awareness of the PSAP operator, meaning that (i) the SIP SUBSCRIBE/NOTIFY framework has been fully integrated into the existing Linphone implementation and (ii) SIP NOTIFY messages carrying sensor data in SensorML format are periodically delivered to the PSAP in order to provide critical data in real-time. Thus, the scope of a realistic functional testing procedure should be primarily to determine the impact that multiple SIP SUBSCRIBE/NOTIFY messages have on behavior of the SIP Core, particularly due to the stochastic creation time of NOTIFY messages, as well as the burstness and volume of the sensor data traffic.

The testing sequence for measuring the number of unacknowledged SIP SUBSCRIBE messages is based on utilizing the default SIP retransmission mechanism over a noisy channel characterised imposing packet loss to the transmitted packets. Towards this end, NetEm was the software of choice as it is an enhancement of the Linux traffic control facilities that allows to add delay, packet loss, duplication and other characteristics to packets outgoing from a selected network interface. Moreover, NetEm is built using the existing Quality Of Service (QoS) and Differentiated Services (diffserv) facilities in the Linux kernel.

Table 3 presents the average number of unacknowledged SIP SUBSCRIBE messages for all the different packet loss used in the test sequence, as well as the corresponding number of retransmissions. For this particular case, the SIP retransmission mechanism was used with the default configuration. Hence, without the loss of generality, the retransmission time (T1) was set to 500 ms while the timeout timer (Timer F) was set at 4 s [3].

Deteriorating network conditions directly affect the number of unacknowledged SIP SUBSCRIBES on the Client side. More specifically, when packet loss is set to 10%, there is an average 3% loss of all SIP SUBSCRIBE messages traversing the infrastructure. In addition, one can see a sharp increase to the retransmission number, reaching an average rate of approximately 4.5.

Moving towards the 5G era, such emergency services architectures must evolve into highly-available, reliable and ultra-low latency service delivery platforms, designed to provide real-time voice, video and real-time text services to PSAPs operators, thus supporting a large variety of added-value services to the emergency agencies. This transformation will also affect healthcare services, since they will be rendered capable of efficiently utilizing emergency response mechanisms integrated in next-generation networks to trigger alerts, notify emergency responders and safeguard high-risk population groups when equipped with wearables and simple sensing equipment [27].

### 3.3. Dominant Healthcare Application Framework Use Cases

The efficient design of IoT-augmented health-monitoring frameworks poses significant challenges, mostly due to the large spectrum of the capabilities and attributes of interconnected devices. Simply merging wireless sensor networks with smart gateways, or replacing existing gateways with smartphones to deploy personal health-monitoring systems are overly simplified solutions to a much more complicated problem. It is therefore essential to take one step back and examine some more generic requirements of such platforms. As mentioned in [14], any contemporary healthcare-monitoring platform must be accessible by all stakeholders, offline or in real-time, regardless of potentially compromising circumstances. Prospective fog-bound healthcare-monitoring solutions must allow patient self-monitoring paired with physician off-line monitoring through seamless access to every available dataset regardless its origin, as well as physician online monitoring and patient monitoring within hospitals, daycare centers or generic healthcare infrastructures [31,32]. Any set of requirements can be consequently classified into one of the following four use cases:Patient self-monitoring, involving end-users willing to monitor their own medical condition, for instance when recuperating from an incident or simply as part of a preventive health monitoring routine with inherent support for early identification of potentially alarming metrics.Physician off-line monitoring, targeting doctors using fog-enabled medical grade wearables to monitor their patients. The patients must carry the wearable device at home and then, after a predefined period, return the device to their physician, allowing the physician to acquire the traces and conduct the diagnosis.Physician on-line monitoring, which extends the previous use case and grants to the physician the ability of remotely monitoring all accumulated vital traces through secure and highly available cloud servicesPatient monitoring within healthcare infrastructure, in which the wearable devices are operated by a personal health assistant or professional caregiver, inside an ambulance, hospital or adult daycare center to instantly acquire patient vitals and carry out real-time situation evaluation before storing the data to the cloud services for future reference and analysis.

These use cases could be further benefit from the upcoming integration with 5G that will allow instant notification of every affiliated/pre-authorized entity in case of emergency, thus placing patients in a position of constant observation by medical professionals which will be always aware of their status in real-time.

## 4. Proposed Architecture

To properly satisfy every use-case requirement and take advantage of the improved connectivity and mobility characteristics delivered by 5G, a new, holistic architecture is presented, meticulously designed to deliver ubiquitous next-generation healthcare services. Our design, as shown in Figure 2, splits interconnected entities into three layers, namely: (i) the Core Network Layer, containing cloud data centers and backed network infrastructure, as analyzed in the previous sections for the case of 5G; (ii) the Edge/Fog Layer, which primarily hosts the basic nodes of the proposed architecture; and (iii) the Sensor/End-device Layer, in which metrics from sensing devices are obtained. With regards to the Edge/Fog Layer, it should be stated here that no distinction between the notions of Fog and Edge computing was made, since from the application perspective and without the loss of generality, the two architectural approaches behave in a similar manner. The main components of our proposed solution along with a detailed functionality description are given in the following paragraphs.

### 4.1. Fog Node

The main pillar of the proposed architecture is the Fog Node, a component responsible for providing all functionality related to data handling, analytics, localization services and user authentication. All critical services are handled by specialized, separate modules as follows:

**(i) Computation engine**: The Fog Node incorporates a dedicated module for handling all data-related tasks in real-time, which provides fast and reliable processing of an unbounded number of streams of data collected from IoT devices, smart phones and web services. The computation engine is deliberately designed with emphasis on speed and data resilience, being able to process a large amount of data collected from sensor nodes within just seconds, while maintaining its operational efficiently regardless of the underlying hardware platform. Facilitated by a dedicated Message Broker, a bus responsible for providing the overall inbound message handling functionality, it is possible to ingest significant volumes of data streams considering the actual hardware it operates onto. The demonstrated ability [33] to operate efficiently even over limited hardware resources is of paramount importance for the overall context of Edge Computing, which dictates smaller datacenter deployment closer to the network edge.

**(ii) Analytics Engine**: Datasets collected from the streams of data as well as the output of the continuous processing taking place inside the Computation Engine are easily selected, extracted and processed to support every possible business intelligence on demand. An integrated, dedicated online Analytics Engine allows the organization of large volumes of data and can visualize them from different points of view. It is possible for the service operators to arbitrary define custom queries, visualization schemes and data manipulation algorithms through an integrated API, thus retrieving every useful information fragment identified inside the accumulated datasets.

**(iii) Database Module**: Data entering the Fog Node can be persisted in their original format and associated with the output of the Computation Engine. This allows a constant data correlation of all output with the original source regardless total number of iterations it undergoes. Data streams can be forwarded later to different components for additional processing, temporary or permanent storage; however, there is always a one-on-one analogy of the context they represent. Offline processing of data is facilitated for archiving services or for benchmarking different versions of components, while the overall data retrieval functionality is carried out through an intuitive API which allows all types of command combinations.

**(iv) Authorization/Authentication Module**: A dedicated module is implemented for handling user access to the Fog Node and the resources or data it contains. User authorization as well as data access can be easily managed in real-time down to specific user, device or time of day, thus limiting the overall visibility of users with limited clearance level. Such a service is considered of paramount importance, especially for healthcare applications, where datasets are deemed sensitive enough to be protected by special legislation. Moreover, to further enhance security, communication interfaces throughout the service infrastructure are designed to be compliant with the most updated standards for internet security and message exchange is encrypted using data encryption standards like AES and TLS/SSL technologies. This approach ensures full communication privacy between all platform entities, eliminating data leaks and eavesdropping or man-in-the-middle attacks.

**(v) Localization Service**: Fog Node encompasses a specialized module for handling Localization Services, since its deployment on the network edge involves a certain degree of sensitivity towards roaming users, which may connect to unaffiliated gateways or move outside their direct intervention zone. This entity keeps track of the user’s positioning and notifies other functions which may need to also adopt their behaviour accordingly.

**(vi) REST API**: To further increase adoption and allow developers to use the overall platform functionality to build applications and interact with its components, the Fog Node exposes a REST API, handled by a dedicated module. Upon receiving proper authorization, developers will be able to make requests to and obtain responses from the core components of the Fog Node, thus initiating automated synergies between their application and the overall system.

### 4.2. Gateway

The Gateway is designed to act a generic communication entity between the three Layers of the architectural paradigm on which next-generation response-capable emergency healthcare services will be deployed. It consists of two different internal nodes, accommodating Fog-to-End-device and Fog-to-Core-Network-Layer packet forwarding; the Fog/Endpoint Gateway and Fog/Cloud Gateway, respectively. These internal nodes can interact with each other as well as communicate with the Fog Node through a customized interface. Multiplexing and data encapsulation techniques, paired with advanced cross-layer optimization methods must be integrated, into creating a sturdy and highly expandable information transmission node, capable of seamlessly handling all types of connections regardless of their origin or destination.

### 4.3. Delivering Response-Capable Emergency Healthcare Services over 5G Infrastructure

By deploying the proposed architecture for delivering response capable, emergency healthcare services over 5G infrastructure, new features are introduced to the overall ecosystem, which significantly boost its effectiveness.

All triggering events start from the end users, which are now equipped with interconnected devices such as Wearable ECG sensors, smartwatches and smartphones. These devices create an intelligent sensing network that obtains metrics related to user vitals and forwards them to the nearby Fog Node for further processing and temporary storage through the affiliated Gateway (Figure 3, pointer 1). When receiving the necessary datasets, the Fog Node can conduct a series of checks against historical or previously stored data that may reveal anomalies, indicating perilous situations. An alert is then automatically triggered by the Fog Node (Figure 3, pointer 2), that initiates an emergency call via the end user’s smartphone (Figure 3, pointer 3). The call traverses the Edge/Fog Layer, and reaches the Core Network Layer through the Network Service Provider infrastructure. It is then further forwarded to the backend Cloud Data Centers (Figure 3, pointer 4), thus reaching the Healthcare Service Providers that are engaged and issue the most appropriate response according the severity of the situation (Figure 3, pointer 5). The response spectrum can be significantly wide, ranging from dispatching an ambulance to simply reset the system to its original state. Nevertheless, it is the proper authority that becomes aware of the actual situation in real-time, and acts on the spot.

Moreover, unlike current emergency healthcare solutions, the proposed architecture provides a much broader set of information such as real-time updates of the user’s condition, accurate location of the emergency incident and recent patient records that greatly help doctors and physicians to prepare in the most accurate way for the specific medical incident. Delivering similar services in the pre-5G Era was simply impossible; however, to make it a reality even with today’s networking features, we need to optimise the cooperation of the network’s building blocks, which is exactly what the proposed architecture intends to do: define a method of interoperability across services, entities, endpoints and key-stakeholders for delivering a seamless end-to-end experience and save lives through accurate and robust data handling.

## 5. Conclusions

As we dive into the 5G and B5G era, highly intelligent, agile and easily programmable network topologies emerge, allowing the deployment of significantly improved services and applications. Furthermore, the healthcare domain is augmented by the proliferation of medical-grade, interconnected wearables that allow highly accurate, real-time monitoring of vital traces and instant comparison of obtained data with vast medical records. The combination of these two fields, through a meticulously designed framework capable of triggering alerts, notifying healthcare service providers or dispatching emergency response units to save lives, is without a doubt a significant upgrade to the quality of life of all individuals. This paper shows how response-capable healthcare services can benefit from the upcoming telecommunication ecosystem disruption, and present, validated and discuss a novel, cutting-edge architecture which integrates new services on top of existing core functionality thus adding significant value through the use cases it supports. Delivering a holistic, end-to-end framework for interconnecting all major health-case service stakeholders is the ultimate goal, and building such a solution by taking advantage of key architectural elements of the 5G ecosystem is as radical as it gets. As future work, we propose the validation of the proof-of-concept in a real B5G testbed that will show that our concept is valid in enabling, in near-zero time, a real emergency communication service.

## Figures and Tables

**Figure 1 sensors-22-03375-f001:**
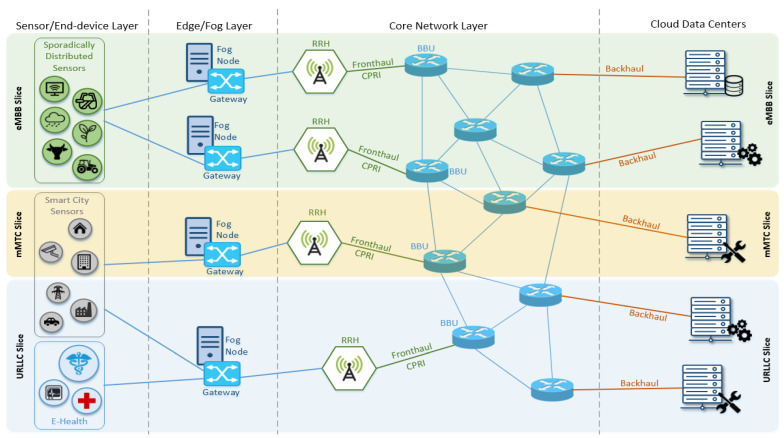
Illustrating the core concept of Network Slicing.

**Figure 2 sensors-22-03375-f002:**
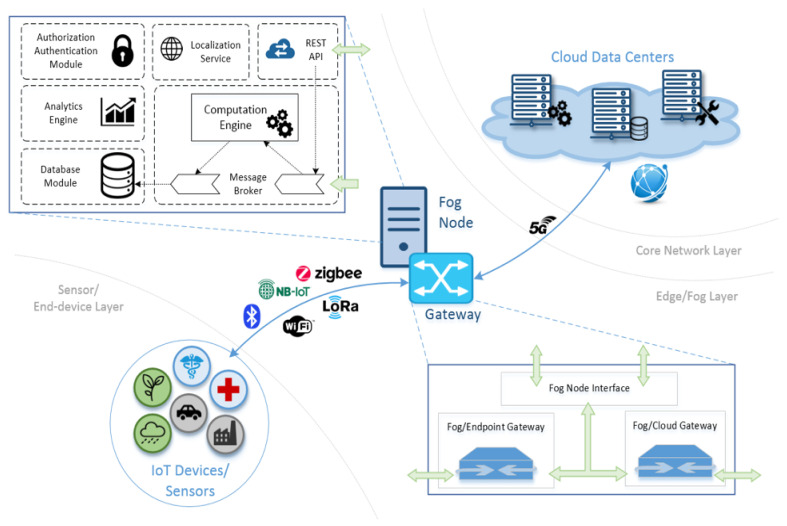
High-level architecture of a generic Edge/Fog Computing node.

**Figure 3 sensors-22-03375-f003:**
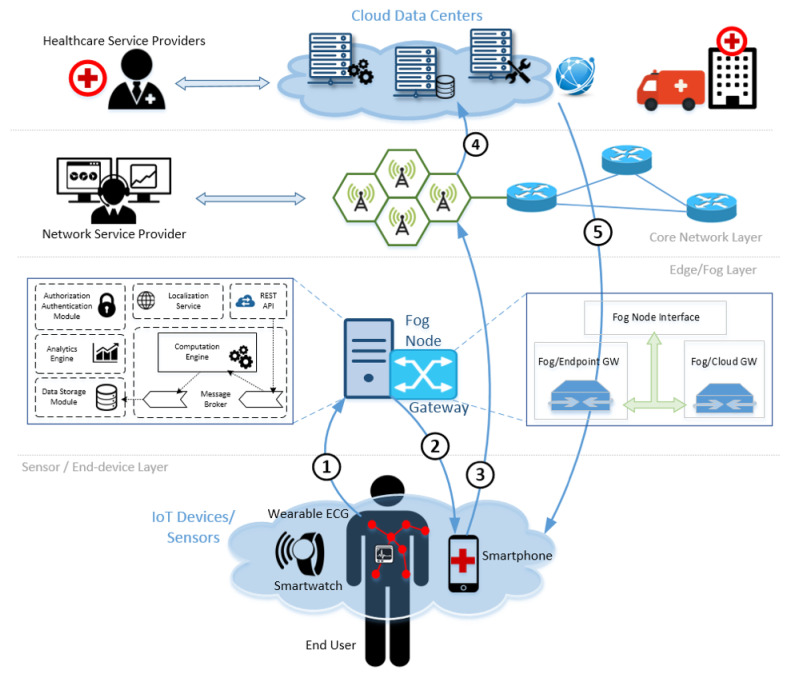
Delivering response-capable emergency healthcare services over 5G infrastructure.

**Table 1 sensors-22-03375-t001:** Average number of ineffective sensor-stemmed SIP MESSAGE.

SIP MSG. Rate (Messages/s)	Transmitted	Successful	Unacknowledged	Re-Transmitted
30	1000	943	57	1539
40	1000	763	237	2152
50	1000	664	336	2305
60	1000	548	452	2424

**Table 2 sensors-22-03375-t002:** Average number of unacknowledged SIP NOTIFY messages.

Rate (Messages/s)	Transmitted	Successful	Unacknowledged	Re-Transmitted
30	1000	975	25	998
35	1000	762	238	1867
40	1000	763	237	2058
45	1000	616	384	2239
50	1000	666	334	2204
55	1000	511	489	2375
60	1000	539	461	2353
30	2000	1677	373	3615
35	2000	1417	583	4305
40	2000	1483	517	4375
45	2000	1173	827	4686
50	2000	1226	774	4649
55	2000	1005	995	4886
60	2000	1061	939	4834

**Table 3 sensors-22-03375-t003:** Average number of ineffective PSAP side-stemmed SIP SUBSCRIBE messages.

Packet Loss (%)	Transmitted	Successful	Unacknowledged	Re-Transmitted
2	1000	986	14	2712
5	1000	985	15	2828
7	1000	984	16	3094
10	1000	970	30	4599

## Data Availability

Not applicable.

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
