# Peer review of "Melding Fog Computing and IoT for Deploying Secure, Response-Capable Healthcare Services in 5G and Beyond"

_sensors, 2022, doi:10.3390/s22093375_

Round 1
Reviewer 1 Report
The paper discussed some concepts and technologies in the 5G era, and the potential architecture to utilize these technologies for response-capable healthcare services. Overall, the paper provided some insights about 5G technologies and how to use these technologies. However, the entire logic seems to lack enough evidence and argumentation.
- There is no simulation, implementation or theoretical analysis of the proposed architecture. How do you validate the proposed architecture?
- There is no comprehensive comparison of the proposed architecture with the previous architecture or other architecture to argue the special architecture design and its novelty.
- Section 2 just introduces different concepts and technologies without explaining how do they connect with the proposed architecture in the paper. They seem not tightly connected with the next parts of the paper.
- What's the purpose of Section 3? Is it about the requirement analysis of response-capable healthcare services? The testing and experiments in Section 3.1 are not tightly connected with other parts of the paper. The reasons for these testing and experiments are not well explained to support the contribution of the research. The experiment results are also not well presented with figures and comparisons.
- The requirement analysis of the use cases is too simple. All the difficulties and issues are not well addressed to support the proposed architecture. Why do these use cases need 5G? Why do we need to have a new architecture to support these use cases?
Reviewer 2 Report
Authors present the architectural elements of 5G for remote healthcare services along with emergency health monitoring. They also propose a holistic scheme based on IoT and Fog Computing.
The paper lacks of a related work section. Authors should review the relateed literature and detail existing published related works such as:
- An architecture and protocol for smart continuous eHealth monitoring using 5G, Computer Networks 129, 340-351. 2017
- Smart system for children's chronic illness monitoring, Information Fusion 40, 76-86. 2018
There are many issues that must be included in the paper in order to accept it:
- There is no performance test of the proposed system
- Conclusion section is missing
- Authors should include their future work at the end of the conclusion section.
Reviewer 3 Report
- Include at the end of the abstract the percentage of improvement of the proposed method.
2. Innovation should be highlighted in the Introduction, as the authors have generally discussed this aspect of the manuscript.
3. In the introduction, you should summarize the manuscript's goals.
4. A table of comparisons should appear in the Literature section. - At least two other methods should be compared in the simulation section. A detailed description of the allocation of service requesters to service providers should be provided.
- Does the paper demonstrate an adequate understanding of relevant literature in the field and cite an appropriate range of literature sources? It seems that a few related works are included in the paper. More recently published papers should be discussed in related work section: Enforcing position-based confidentiality with machine learning paradigm through mobile edge computing in real-time industrial informatics.
- In the conclusion, the practical application field of the proposed methods and the research findings can be described that highlights the contribution of this article. Then, the advantages (and disadvantages?) of the proposed methods should be discussed.
- Figures and tables that are about experimental results, require more explanation.
- The results presented need more clarity in their description. The diagrammatic representation seems fine, but they need to be presented in more detail in the article. This makes things easier for the reader to understand.
- Results and discussion section: The presentation of results should be simple and straightforward in style. You should improve your analyzing and also present the comparison between performance of your approach and other researches. Results given in figures should not be repeated in tables.
Author Response
"Please see the attachment."

Round 2
Reviewer 1 Report
Acceptable.
Reviewer 2 Report
Authors have fixed all my comments
Reviewer 3 Report
Revised properly